# Oncogenic mutation or overexpression of oncogenic KRAS or BRAF is not sufficient to confer oncogene addiction

**Reina E. Ito[1,2], Chitose Oneyama[3], Kazuhiro Aoki [1,2,4] ***

**1** Quantitative Biology Research Group, Exploratory Research Center on Life and Living Systems (ExCELLS), National Institutes of Natural Sciences, Okazaki, Aichi, Japan, **2** Division of Quantitative Biology, National Institute for Basic Biology, National Institutes of Natural Sciences, Okazaki, Aichi, Japan, **3** Division of Cancer Cell Regulation, Aichi Cancer Center Research Institute, Nagoya, Aichi, Japan, **4** Department of Basic Biology, School of Life Science, SOKENDAI (The Graduate University for Advanced Studies), Okazaki, Aichi, Japan

* k-aoki@nibb.ac.jp

**Data Availability Statement:** All relevant data are within the manuscript and its Supporting Information files.

**Funding:** K.A. is supported by JST, CREST Grant No. JPMJCR1654; and by MEXT/JSPS KAKENHI

## Abstract

Oncogene addiction is a cellular property by which cancer cells become highly dependent on the expression of oncogenes for their survival. Oncogene addiction can be exploited to design molecularly targeted drugs that kill only cancer cells by inhibiting the specific oncogenes. Genes and cell lines exhibiting oncogene addiction, as well as the mechanisms by which cell death is induced when addicted oncogenes are suppressed, have been extensively studied. However, it is still not fully understood how oncogene addiction is acquired in cancer cells. Here, we take a synthetic biology approach to investigate whether oncogenic mutation or oncogene expression suffices to confer the property of oncogene addiction to cancer cells. We employed human mammary epithelium-derived MCF-10A cells expressing the oncogenic KRAS or BRAF. MCF-10A cells harboring an oncogenic mutation in a single-allele of KRAS or BRAF showed weak transformation activity, but no characteristics of oncogene addiction. MCF-10A cells overexpressing oncogenic KRAS demonstrated the transformation activity, but MCF-10A cells overexpressing oncogenic BRAF did not. Neither cell line exhibited any oncogene addiction properties. These results indicate that the introduction of oncogenic mutation or the overexpression of oncogenes is not sufficient for cells to acquire oncogene addiction, and that oncogene addiction is not associated with transformation activity.

## Introduction

Most human cancers develop over a long period of time, from a few years to several decades, as mutations accumulate in various proto-oncogenes and tumor suppressor genes [1, 2]. During this process, cancer cells rewire the intracellular signal transduction system by accumulating mutations and epigenetic changes, and consequently acquire the characteristics of malignant tumors (Fig 1). On the other hand, it is well-established that the overexpression of

Grant No. 16KT0069, 16H01425 "Resonance Bio", 18H04754 "Resonance Bio", 18H02444, and 19H05798. The funders had no role in study design, data collection and analysis, decision to publish, or preparation of the manuscript.

**Competing interests:** The authors have declared that no competing interests exist.

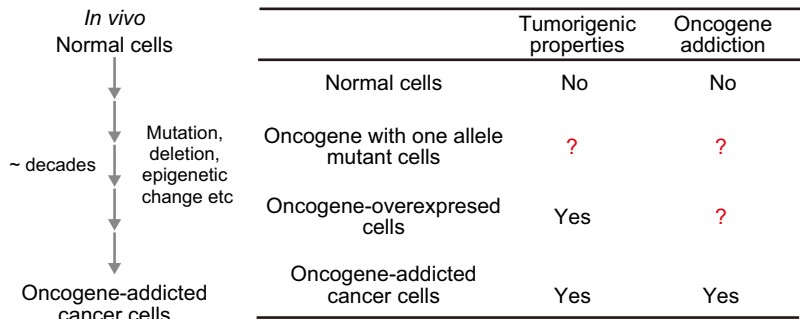

| | Tumorigenic properties | Oncogene addiction |
|---|---|---|
| Normal cells | No | No |
| Oncogene with one allele mutant cells | ? | ? |
| Oncogene-overexpressed cells | Yes | ? |
| Oncogene-addicted cancer cells | Yes | Yes |

**Fig 1. Tumorigenic properties and oncogene addiction.** (A) *In vivo* process by which cancer cells acquire oncogene addiction. (B) The table compares different types of cells with respect to tumorigenic properties and oncogene addiction.

oncogenes suffices for the neoplastic transformation of non-cancerous cells *in vitro* and *in vivo*, resulting in infinite proliferation, anchorage independence, and angiogenesis [3–5]. Therefore, properties that can be acquired over a long period of time appear to be different from the tumorigenesis induced by the proto-oncogenes/tumor suppressor genes activation/ inactivation.

Oncogene addiction (or oncogene pathway addiction) is a characteristic of cancer cells in which malignant cells are dependent for their proliferation and survival on a particular proto-oncogene and/or tumor suppressor gene [6, 7]. Thus, the proliferation and survival of oncogene-addicted cancer cells are dramatically impaired by suppression of the oncogenes. For example, the inhibition of addicted oncogenes with RNAi or small chemical inhibitors causes apoptosis in oncogene-addicted cancer cells, but not in other cells, thereby providing a rationale for molecularly targeted therapy [8]. Imatinib (Gleevec), a BCR-ABL1 kinase inhibitor, and Gefitinib (Iressa), an EGFR inhibitor, are typical examples of drugs successfully targeted to the appropriate molecules and are effective for the treatment of chronic myeloid leukemia (CML) and non-small cell lung cancer, respectively [9]. Several molecular mechanisms by which cancer cells die through acute inhibition of addicting oncogenes selectively required for survival have been reported, including oncogene shock, oncogene amnesia, genetic streaming, synthetic lethality, and others [10, 11]. However, little is known about how and when the property of oncogene addiction is acquired, and which oncogene(s) is prone to cause oncogene addiction, although the phenomenon has been reported to involve epigenetic DNA changes that accompany the development of cancer [12].

The Ras-ERK signaling pathway plays a pivotal role in a wide range of cell functions such as cell proliferation, differentiation, and survival, but also plays a key role in tumorigenesis [13, 14]. Indeed, the *KRAS* gene is the second-most frequently mutated gene in human cancers, after the *p53* gene, and the *BRAF* gene is also frequently mutated in melanoma and colorectal cancer [2]. KRAS- or BRAF-mutated cancer cells also exhibit oncogene addiction. Suppression of the expression of mutated KRAS by antisense or siRNA caused cell cycle arrest and apoptosis in KRAS-mutated cultured cancer cell lines, and epithelial-mesenchymal transition (EMT) was closely associated with KRAS dependency [15, 16]. Knockdown by RNAi or treatment with a BRAF-selective inhibitor leads to the inhibition of cell proliferation and survival in BRAF-mutated cancer cell lines [17–20]. However, these results were obtained by using cell lines established from human patients, it is impossible to trace when and how oncogene addiction is acquired. Interestingly, the expression of oncogenic HRAS or KRAS has been shown to induce tumor formation *in vivo* in a doxycycline-dependent manner, and withdrawal of the

drug resulted in tumor shrinkage [21, 22]. However, Chin et al. also showed that these cells do not alter their growth rate regardless of doxycycline treatment *in vitro* [21]. Thus, it remains unclear whether oncogene addiction is achieved by the acquisition of tumorigenic properties through expression of the *KRAS* or *BRAF* oncogenes. In this study, we examined whether an oncogenic mutation in a single allele of *KRAS* or *BRAF* or overexpression of *KRAS* or *BRAF* oncogenes was sufficient to induce oncogene addiction.

## Materials and methods

### Plasmids, reagents, and antibodies

The plasmids made in this study are listed as follows: pCSIIneo-MCS (multi-cloning site), pCSIIneo-FLAG-BRAF-V600E, pCSIIneo-FLAG-KRAS-G12V, psPAX2 was a gift from Dr. D. Trono, Addgene plasmid #12260) [23]. pCMV-VSV-G-RSV-Rev (a gift of Dr. Miyoshi, RIKEN, Japan). pCSIIbsr-FLAG-BRAF-V600E was a kind gift from Dr. Matsuda (Kyoto University) [24].

The antibodies used for western blot and immunofluorescence analyses are as follows: phospho-anti-Erk1/2 (Thr202/Tyr204) (E10) and anti-Erk1/2 (137F5) were from Cell Signaling Technology; anti-KRAS (clone 3B10-2F2) was from Sigma; anti-Raf-B (F-7) and anti-Tubulin (sc-58886) were from Santa Cruz Biotechnology.

The negative control siPOOL, BRAF targeted siPOOL and KRAS targeted siPOOL were purchased from siTOOLs Biotech.

### Cell lines

The A549, H358 (CI-H358), and A375 cell lines were purchased from the American Type Culture Collection. Lenti-X 293T cells were purchased from the Invitrogen. The 293T cells were maintained in DMEM high glucose (Cat#08459–64, Nacalai Tesque) supplemented with 10% fetal bovine serum. H358 cell lines were maintained in RPMI 1640 media (ATCC modification) (Cat#A10491-01, Gibco) supplemented with 10% fetal bovine serum. The A549 and A375 cell lines were maintained in DMEM high glucose (Cat#08459–64, Nacalai Tesque) supplemented with 10% fetal bovine serum and in DMEM high glucose (Cat#08459–64, Nacalai Tesque) supplemented with sodium pyruvate and 10% fetal bovine serum, respectively. All cell lines were maintained at 37˚C under 5% $CO_2$.

MCF-10A parental cells, MCF-10A BRAF V600E/+ cells, and MCF-10A KRAS G12V/+ cells (catalog numbers HD PAR-003, HD101-012, and HD101-004) were purchased from Horizon Discovery. KRAS G12V OE and BRAF V600E OE were established through lentivirus-mediated gene transfer into the parental MCF-10A cells. In brief, the lentiviral pCSIIneo or pCSIIbsr vectors were transfected into Lenti-X 293T cells (Clontech) together with the packaging plasmid psPAX2, and pCMV-VSV-G-RSV-Rev by using the linear polyethyleneimine "Max" MW 40,000 (Polyscience). After two days, MCF-10A parental cells were cultured in the virus-containing media in the presence or absence of 8 μg/mL polybrene for 3–4 hrs. Two days after infection, the cells were selected by at least one-week treatment with 150 ug/ml G418 or 10 μg/mL blasticidin (InvivoGen, San Diego, CA). Bulk populations of selected cells were used in this study. An empty vector, pCSIIneo-MCS, was used as a control. All cell lines were maintained at 37˚C under 5% $CO_2$ with antibiotics.

MCF-10A cell lines were maintained in the full growth medium, which consisted of DMEM/F12 (1:1) (Cat#11330–032, Gibco) supplemented with 5% horse serum (Cat#16050–122, Invitrogen), 10 mg/ml insulin (Cat#12878–44, Nacalai Tesque), 0.5 mg/ml hydrocortisone (Cat#1H-0888, Invitrogen), 100 ng/ml cholera toxin (Cat#101B, List Biological Laboratories), 20 ng/ml hEGF (Cat#AF-100-15, PeproTech), and 1% penicillin/streptomycin (Cat#26253–84,

Nacalai Tesque). For some experiments, partial growth medium and starvation medium were used; the former contained DMEM/F12 (1:1) supplemented with 5% horse serum and 1% penicillin/streptomycin, and the latter consisted of DMEM/F12 (1:1) supplemented with 2% horse serum, 10 mg/ml insulin, 0.5 mg/ml hydrocortisone, 100 ng/ml cholera toxin, and 1% penicillin/streptomycin.

## Genomic DNA preparation and sequencing

Genomic DNA was prepared from cells using QuickExtract solution (Nalgene) following the manufacturer's instruction. PCR amplification was done using KOD FX Neo (Toyobo). PCR primers to amplify DNA were designed to target the oncogene mutated region (15th exon for *BRAF* and 2nd exon for *KRAS*). Direct sequencing of PCR products was carried out by FASMAC. The amplification primers were as follows: BRAF-Fw, 5'-ATCTCACCTCATCCTAA CACATTTCAAGCCCC-3'; BRAF-Rv, 5'-GACTTTCTAGTAACTCAGCAGCATCTCAGGGC C-3'; KRAS -Fw, 5'-GCCTGCTGAAAATGACTGAA-3'; KRAS-Rv, 5'-AGAATGGTCCT GCACCAGTAA-3'.

## Soft-agar colony formation assay

A series of MCF-10A cells (2 x 10^4 per well) were mixed with 0.3% agarose, low gelling temperature (Cat#35640, SIGMA) in the full growth medium, plated on top of a solidified layer of 0.6% agarose in full growth medium [25] in a 6-well plate, and fed every 3 days with full growth medium. Photographs were taken by an OLYMPUS CKX53 inverted microscope with a DP20 digital camera (OLYMPUS). Finally, the colonies were stained with MTT (1 mg/ml in PBS solution) and imaged using an EPSON GT-X900 scanner. The images were analyzed with ImageJ (Fiji), extracting the number of colonies that exceeded a certain threshold intensity. When we combined soft agar assay with the siRNA experiment, 1 nM siRNA-treated cells were embedded in the soft agar, fed only one week later and cultured for 2 weeks. MCF-10A cell lines were transfected with 1 nM siPOOLs, maintained for 2 days in partial growth medium, and then 2 x 10^4 cells were embedded in full growth medium-based soft agar. In the case of cancer-derived cell lines, cells were first transfected with 1 nM siPOOLs in RPMI supplemented with 10% FBS. One day after transfection, the cells were embedded in the soft agar based on RPMI supplemented with 10% FBS. The cell number embedded in the soft agar is as follows: A549, 2 x 10^4 cells; H358, 4 x 10^4 cells; A375, 1 x 10^4 cells.

## Western blot analysis

Cells were washed once with PBS and lysed directly in 1x SDS sample buffer (1 M Tris-HCl pH 6.8, 50% glycerol, 10% SDS, 0.2% bromophenol blue, and 10% 2-mercaptoethanol). When the number of cells decreased due to knock-down of *KRAS* or *BRAF*, the volume of SDS sample buffer was reduced for cell lysis. After sonication and heat denaturation by boiling, the samples were separated by premade 5–20% gradient SDS-polyacrylamide gel electrophoresis (PAGE) (Nakalai or Atto) and transferred to Immobilon-FL Polyvinylidene Difluoride (PVDF) membranes (Millipore, Billerica, MA). After blocking with skim milk (Morinaga, Tokyo) or Odyssey blocking buffer (LI-COR), the membranes were incubated with primary antibodies diluted in skim milk, BSA, or Odyssey blocking buffer (LI-COR), followed by secondary antibodies diluted in Odyssey blocking buffer. Proteins were then detected by an Odyssey Infrared scanner (LI-COR) and analyzed by using the Odyssey software. The detection conditions are as follows: Resolution, 168 μm (Figs 2F and 5F KRAS and BRAF, and S3C, S4A and S5C Figs) or 84 μm (Fig 5F empty vector, and S1D, S2A, S2C, S3A and S5A Figs); sensitivity (scanning speed), normal (Figs 2F and 5F KRAS and BRAF, and

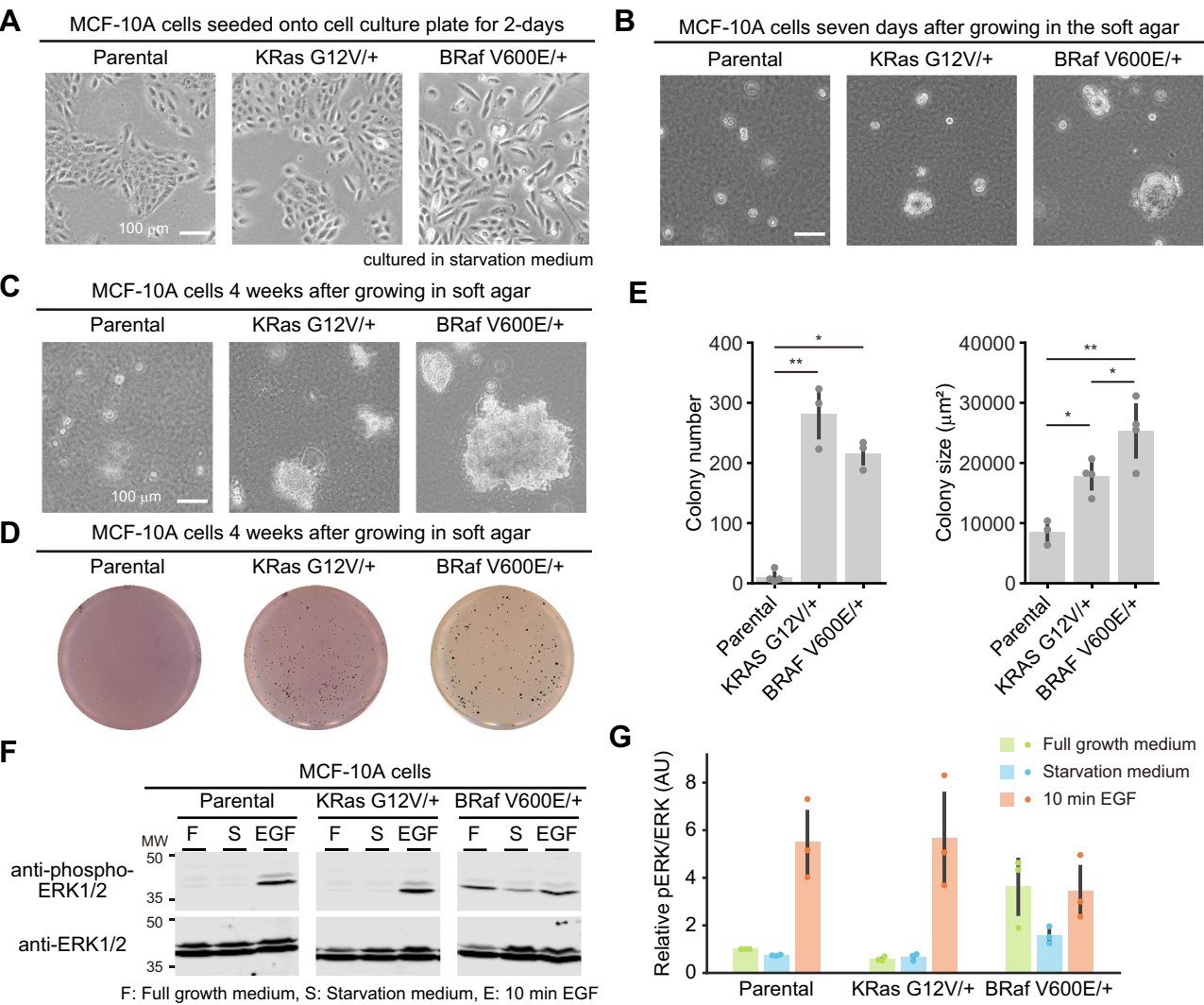

**Fig 2. Characterization of MCF-10A cells harboring a single allele mutation of KRAS G12V or BRAF V600E.** (A and B) Morphology of the indicated MCF-10A cells seeded onto a cell culture dish for two days (A) or seeded in soft agar for seven days (B). (C) Morphology of the indicated MCF-10A cells seeded in soft agar for 4 weeks. (D) Representative images of MTT-stained colonies of the indicated MCF-10A cells seeded in soft agar for 4 weeks. (E) The mean number of colonies (left) and the mean colony size ($\mu m^2$) (right) are shown with the SD. Dots indicate actual data points. The numbers of experiments are as follows: Parental, n = 4 (left) and 3 (right); KRAS G12V/+, n = 3 and 4; BRAF V600E/+, n = 3 and 4. (F) Western blot analysis from the parental MCF-10A, KRAS G12V/+, and BRAF V600E/+ cell lines under the indicated conditions. (G) Quantification of ERK phosphorylation in panel F. Relative pERK/ERK values normalized by parental MCF-10A cells cultured in full growth medium condition are shown with the SD (n = 3). Dots indicate actual data points. The results of statistical analysis are shown in S1B Fig. * $p < 0.05$, ** $p < 0.01$.

S3C, S4A and S5C Figs) or low (Fig 5F empty vector, and S1D, S2A, S2C, S3A and S5A Figs). For the analysis of ERK phosphorylation in the parental MCF-10A, KRAS G12V/+, and BRAF V600E/+ cell lines, cells were seeded with full growth medium for 1 day before the serum starvation; washed once with PBS and changed to starvation medium for 24 hrs; and finally treated with 10 ng/ml EGF for 10 min. In the case of ERK phosphorylation analysis in MCF-10A cells overexpressing KRAS G12V or BRAF V600E, the MCF-10A cells were seeded with full growth medium or starvation medium, cultured for 24 hours, and treated with 10 ng/ml EGF for 10 min. For the quantification, the intensity of ERK1/2 or Tubulin signal was used as a loading control.

## Crystal violet staining

For siRNA-mediated knockdown experiments to examine oncogene-addiction, we used siPOOLs. Reverse transfection of cancer cell lines and MCF-10A cell lines with siPOOLs was performed in Nunc edge 2.0 (96-Well Plates, Thermo Fisher) using Lipofectamine RNAiMax reagent (Invitrogen) according to the siPOOLs transfection protocol. Seeding density and siPOOL concentrations were modified depending on the cell lines and culture conditions used. After 3 to 4 days of siPOOL treatment, cells were washed once with PBS and stained with 0.1% crystal violet for 10 min, then washed three times with PBS and photographed with an EPSON GT-X900 scanner (1200 dpi). Relative cell number calculations were performed by ImageJ (Fiji) in comparison with the corresponding cell lines treated with a negative control siPOOL. For cancer cell lines, 1 nM siPOOL-reverse transfected cells were cultured in the RPMI 1640 medium (ATCC modification) (Cat#A10491-01, Gibco) supplemented with 10% fetal bovine serum for 3–4 days with the following cell densities, 6x10^3 cells/well for H358, 1x10^3 cells/well for A549, and 1-2x10^3 cells/well for A375. And knock-down experiment of the derivative cell lines of MCF-10A was performed the following conditions; in partial growth medium condition, reverse transfection in 6x10^3 cells with 1 nM siPOOLs for 3–4 days, and in starvation medium condition, reverse transfection in 3x10^3 cells with 0.5 nM siPOOLs for 3–4 days.

## Statistical analysis

In Figs 1–6 and S1 Fig, the Kruskal–Wallis test, a non-parametric alternative to the one-way ANOVA, was adopted, because the data points are non-normally distributed. When $p$-value of the Kruskal-Wallis test was less than 0.05, as a post-hoc analysis the Conover–Iman test with the Bonferroni–Holm Correction for multiple testing was adopted to calculate $p$-values. These statistical tests were conducted on Python 3 and SciPy. In S2, S3 and S5 Figs, a paired $t$-test was adopted with the Microsoft excel software.

## Results

### *In vitro* characterization of MCF-10A cells harboring a single allele mutation of KRAS G12V or BRAF V600E

The human normal mammary gland-derived MCF-10A cell lines were used in this study. This is because MCF-10A was spontaneously immortalized without defined factors [26], the cell line is not tumorigenic, *i.e.*, they are not able to grow under anchorage-independent conditions or to form tumors when injected subcutaneously into nude mice [27], it does not have mutations in *KRAS* and *BRAF*, and it is easy to culture. In addition, it is beneficial to use MCF-10A because of the availability of cell lines with KRAS G12V and BRAF V600E as we mention below. On the other hand, it has been reported that MCF10A lacks a tumor suppressor gene, *p16*, which may render the cell line immortalized. Therefore, it should be noted that it is substantially different from normal mammary epithelial cells (see Discussion).

To reconstitute oncogene addiction, we first obtained MCF-10A cells harboring KRAS G12V or BRAF V600E mutation, which were generated by genome editing with adeno-associated virus (hereafter referred to as KRAS G12V/+ or BRAF V600E/+ cells) [27, 28]. Mutation of a single allele of KRAS G12V and BRAF V600E was confirmed by direct sequencing (S1A Fig). KRAS G12V/+ cells grew as efficiently as parental MCF-10A cells onto two-dimensional dishes, showing islands of densely packed cells (Fig 2A). Meanwhile, BRAF V600E/+ cells exhibited more scattered and fibroblastic morphology (Fig 2A). We next evaluated anchorage-independent colony formation in soft agar, which is a feature of transformation [29]. Seven

days after seeding in the soft agar, the parental MCF-10A cells hardly proliferated, whereas KRAS G12V/+ cells grew slowly and formed small spheroids (Fig 2B). This result is not consistent with the previous report using the same cells [28], possibly because of the difference of the experimental conditions under which the seeding cell number (3x10^4 cells vs 2x10^4 cells), the top layer agar concentration (0.3% vs 0.4%), the interval of medium addition (every 3 days vs every 1 week), and/or incubation time (4 weeks vs 3 weeks). Under the same condition, BRAF V600E/+ cells formed larger and more spheroids than KRAS G12V/+ cells (Fig 2B) in good agreement with the previous report [27], suggesting an increase in transformation activity in BRAF V600E/+ cells. Despite the smaller size of colonies, the number of colonies in KRAS G12V/+ cells was comparable to that of BRAF V600E/+ cells 4 weeks after seeding in the soft agar (Fig 2C–2E). Consistent with these data, BRAF V600E/+ cells showed significantly higher basal phosphorylation of ERK, downstream of KRAS and BRAF, under normal and serum-starved conditions (Fig 2F and 2G, S1B Fig). Parental and KRAS G12V/+ cells responded well to EGF stimulation, while BRAF V600E/+ cells demonstrated less sensitivity to EGF, probably because of its higher basal activity (Fig 2F and 2G, S1B Fig). The expression levels of KRAS and BRAF showed no substantial changes in these cell lines (S1C–S1E Fig), suggesting that the differences between parental cells and the mutant cell lines were attributable to the increased activity of KRAS and BRAF. These results indicated that a single allele mutation of BRAF V600E in MCF-10A cells enhances transformation activity in culture, while a single allele KRAS G12V results in cells with similar or slightly increased *in vitro* transformation activity.

## Evaluation of oncogenic KRAS or BRAF addiction in MCF-10A cells harboring a single allele mutation of KRAS G12V or BRAF V600E

We next quantified to what extent cells were addicted by the expression of *KRAS* or *BRAF* oncogene. For this purpose, the effect of KRAS or BRAF ablation on cell growth was examined with crystal violet staining and RNA interference (RNAi) [16, 30]. In addition, to reduce the off-target effect of RNAi, siPOOLs were used to deplete KRAS and BRAF; siPOOLs dilute the off target effects by pooling multiple siRNAs against the target genes [31]. The siPOOLs for *KRAS* or *BRAF* do not discriminate between wild-type and mutant sequences of *KRAS* or *BRAF*.

As a control, we used two lung cancer-derived cell lines, A549 (homozygous KRAS G12S mutation; KRAS non-addicted) and H358 (heterozygous KRAS G12C mutation; KRAS-addicted) [16], and melanoma-derived cell line, A375 (homozygous BRAF V600E mutation; BRAF-addicted) [19]. The knockdown of KRAS or BRAF with siPOOLs was confirmed in these cell lines (S2A and S2B Fig). A549 cells showed modest inhibition of cell growth by KRAS depletion (Fig 3A and 3B). H358 and A375 cells exhibited strong suppression of cell growth by KRAS and BRAF ablation with the siPOOLs (siRNA), respectively (Fig 3A and 3B). In A375 cells, knock-down of the BRAF gene caused a concomitant decrease in both pERK and cell proliferation, which was consistent with previous reports showing that cancer cells harboring BRAF V600E are addicted to ERK pathway (S2A and S2B Fig) [32]. In H358 cells cultured in RPMI supplemented with 10% FBS, knock-down of the KRAS gene suppressed cell proliferation but not pERK with statistical significance (S2A and S2B Fig). Meanwhile, we could reproduce that ERK phosphorylation was significantly reduced in H358 cells cultured in 5% FBS medium as previously shown (S2C and S2D Fig) [16]. Importantly, the inhibition of cell growth was observed even under a full growth condition for these cancer cell lines; these cells were indeed addicted to the expression of the KRAS or BRAF oncogene for their cell growth.

We further investigated the effect of *KRAS* or *BRAF* knock-down on anchorage independent growth in these cancer-derived cell lines. The cells were introduced with siPOOLs and embedded in soft agar 1 day after the siPOOLs transfection. Of note, the effect of siPOOLs-mediated knock-down would not last for a long time because of the dilution of siRNA through cell proliferation, and therefore we presume that siPOOLs-mediated knock-down affects cell survival and proliferation in the early stages of the soft agar colony formation. Nevertheless, we found that the number of colonies in A549 cells and H358 cells was significantly reduced by KRAS depletion (Fig 3C and 3D). BRAF depletion decreased the number of colonies in A375 cells and H358 cells (Fig 3C and 3D). These results indicate the involvement of oncogenic KRAS or BRAF in *in vitro* transformation properties.

We then examined whether KRAS G12V/+ and BRAF V600E/+ cells were addicted to their oncogenes. If oncogene addiction has been acquired, we expect that knock-down of the oncogene would result in phenotypes such as apoptosis in oncogene addicted cells, but not in

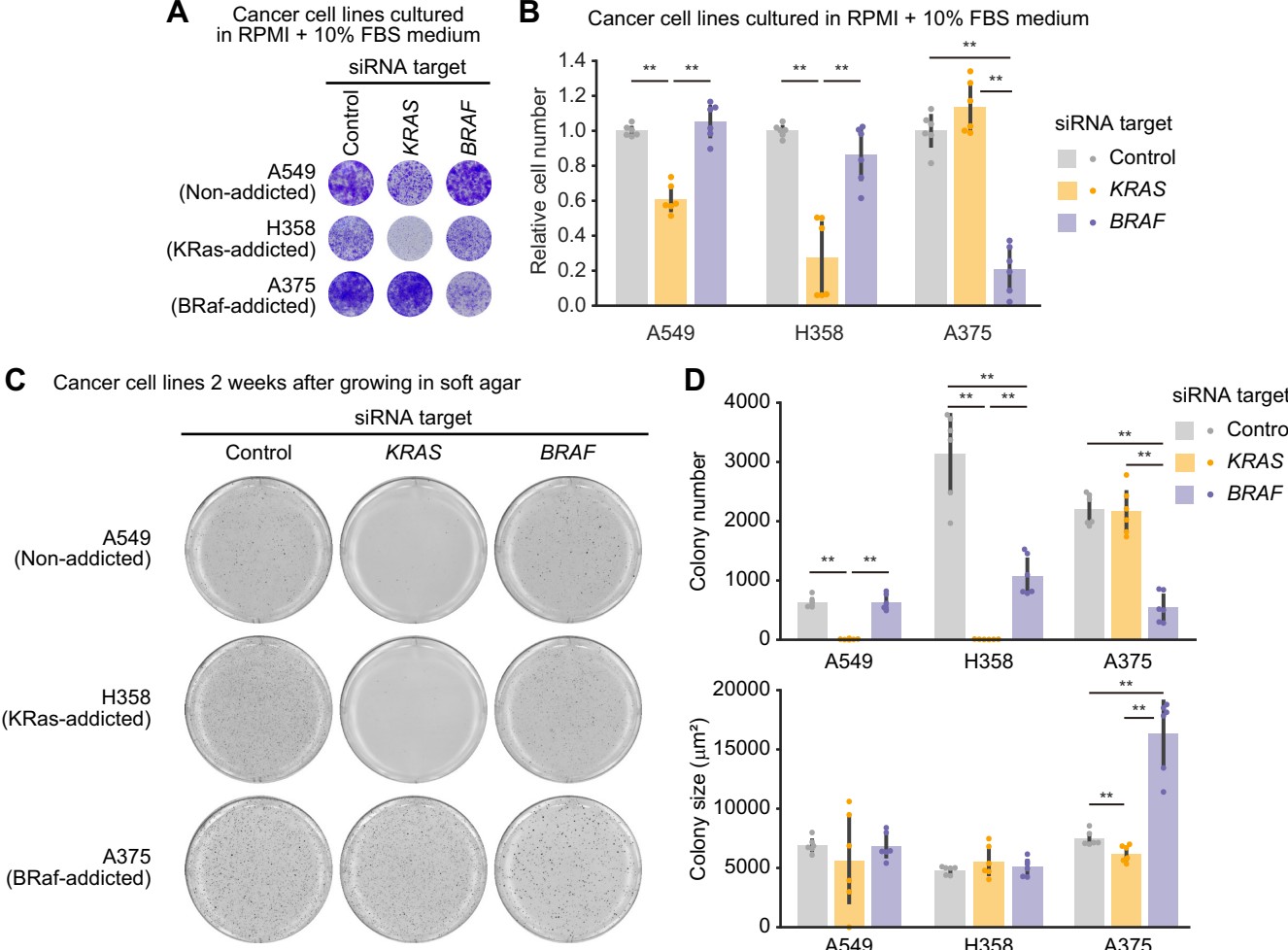

**Fig 3. Evaluation of oncogenic KRAS or BRAF addiction in cancer-derived cell lines.** (A and B) Cell growth assays following siRNA-mediated negative control, KRAS, or BRAF ablation in the A549, H358, and A375 cell lines. Four days after transfection with siRNA, relative cell densities were determined by crystal violet staining. (A) Representative 96-well plates are shown. (B) The mean relative cell number is shown with the SD. Dots indicate actual data points. The numbers of experiments are as follows: A549, n = 6; H358, n = 6; A375, n = 6. (C) Representative images of MTT-stained colonies of the indicated cells seeded in soft agar for 2 weeks. (D) The mean number of colonies (upper) and the mean colony size (µm²)(lower) are shown with the SD. Dots indicate actual data points. The numbers of experiments are as follows: A549, n = 6; H358, n = 6; A375, n = 6. * $p < 0.05$, ** $p < 0.01$.

normal cells and oncogene non-addicted cells. First, siPOOLs were introduced into parental, KRAS G12V/+, and BRAF V600E/+ MCF-10A cells cultured in "full growth medium". However, we were unable to reproducibly quantify the effect of RNAi on KRAS or BRAF ablation and cell growth. This is because the seeding density at the time of cell passage has a great impact on the assay even a few days after the passage. Therefore, we used "partial growth medium", which contained DMEM and serum without the supplements included in the "full growth medium". Under the culture condition with the partial growth medium, MCF-10A cells grew and proliferated slowly, making it easier to assess the depletion of KRAS and BRAF (S3A and S3B Fig). Parental MCF-10A cells did not show substantial change in cell growth by the knock-down of the *KRAS* or *BRAF* gene (Fig 4A and 4B). We found that neither KRAS nor BRAF depletion resulted in substantial effect on cell growth in KRAS G12V/+ and BRAF V600E/+ cells (Fig 4A and 4B), indicating that cell growth in these cells was not dependent on the expression of the *KRAS* or *BRAF* oncogene.

The oncogenic mutation in a single allele of *KRAS* or *BRAF* leads to sustained proliferative signaling [2]. Indeed, the introduction of oncogenic mutation in *KRAS* or *BRAF* enabled MCF-10A cells to grow in the culture medium without EGF and serum, namely, the "starvation medium" (Fig 4C). We confirmed depletion of KRAS and BRAF under the starvation condition (S3C and S3D Fig). Interestingly, the growth factor independence was derived from KRAS expression in KRAS G12V/+ cells, whereas it was not derived from BRAF expression in BRAF V600E/+ cells (Fig 4D and 4E). We then investigated the involvement of oncogenic mutation of *KRAS* or *BRAF* in anchorage-independent growth in KRAS G12V/+ and BRAF V600E/+ cells. KRAS depletion highly reduced the number of colonies in these two cell lines (Fig 4F and 4G). Knock-down of the *BRAF* gene also attenuated the number of colonies in these cell lines, but the effect was stronger in BRAF V600E/+ cells than in KRAS G12V/+ cells (Fig 4F and 4G). Although these results indicate that oncogenic mutation in *KRAS* or *BRAF* is involved in the acquisition of the ability for growth factor-independent and anchorage-independent growth, they are not suitable for evaluation of oncogene addiction because parental cells did not proliferate in the starvation medium and in the soft agar. In sum, we concluded that KRAS G12V/+ cells and BRAF V600E/+ cells were not addicted to the oncogene, even though these cells acquired the ability of growth factor-independent and anchorage-independent growth.

## *In vitro* characterization of MCF-10A cells overexpressing KRAS G12V or BRAF V600E

Next, we examined whether overexpression of KRAS G12V or BRAF V600E induced the property of oncogene addiction. The FLAG-tagged KRAS G12V or BRAF V600E oncogene was introduced into parental MCF-10A cells through lentivirus, producing MCF-10A cells over-expressing KRAS G12V or BRAF V600E (hereinafter referred to as "KRAS G12V OE" or "BRAF V600E OE" cells). As a control, we introduced the empty vector into MCF-10A cells through lentivirus (referred to as "empty vector"). The expression levels of KRAS, BRAF, and ERK were compared with parental cells (S1C–S1E Fig). During the course of experiments, we recognized that long-term culture of BRAF V600E OE cells reduced the expression levels of BRAF V600E and ERK phosphorylation levels (S4A Fig), probably due to the adaptation by a reduction of BRAF V600E expression through gene silencing and/or negative feedback mechanisms [33]. Thus, we referred to early-passage (< 1 week from the establishment of cell lines) and late-passage (> 1 week) cells as BRAF V600E OE early cells and BRAF V600E OE late cells, respectively.

The morphology of KRAS G12V OE cells cultured on the plastic dish was scattered and fibroblastic, whereas BRAF V600E OE late cells exhibited a typical epithelial cell shape to the

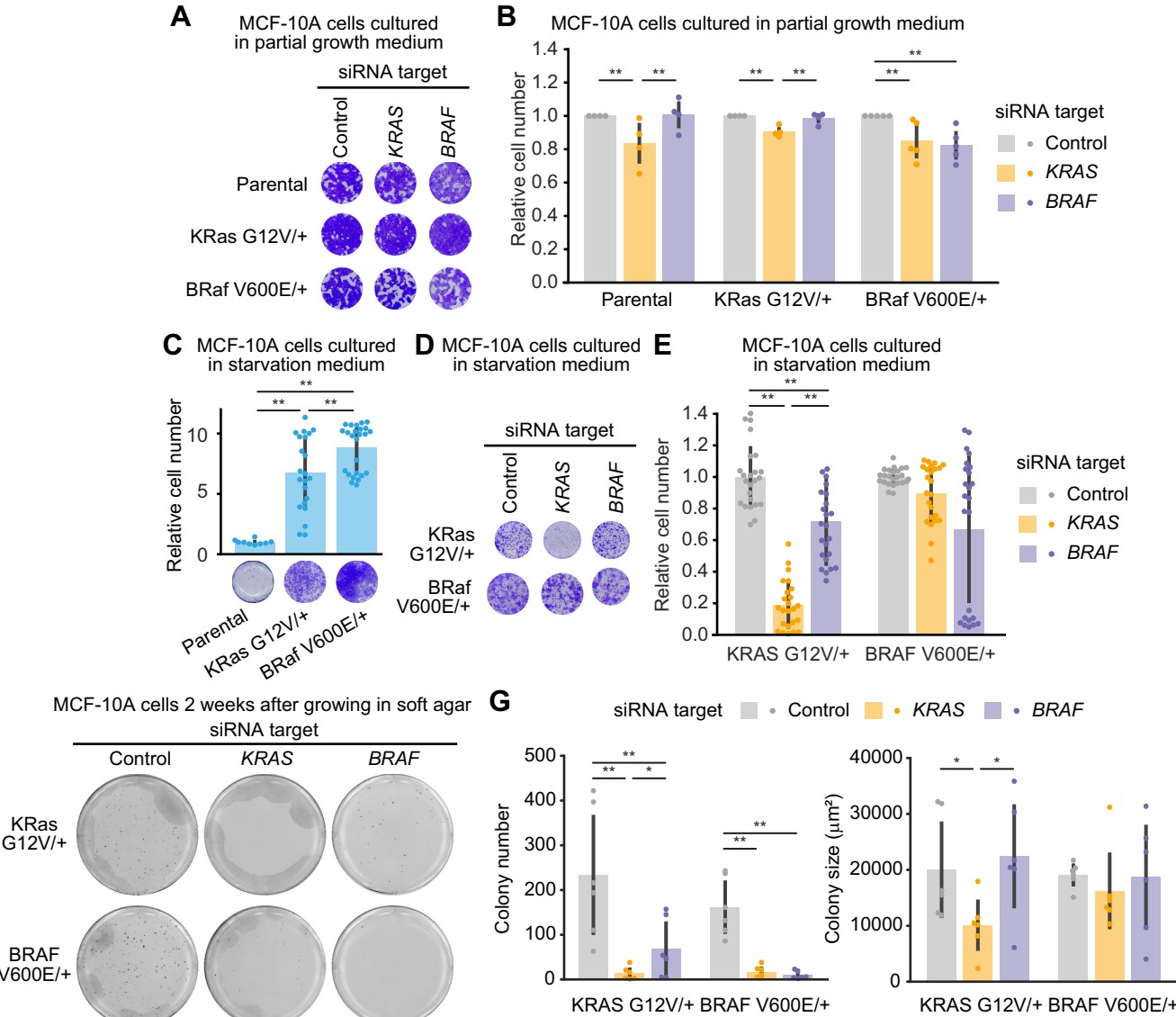

**Fig 4. Evaluation of oncogenic KRAS or BRAF addiction in MCF-10A cells harboring a single allele mutation of KRAS G12V or BRAF V600E.** (A and B) Cell growth assays following siRNA-mediated negative control, KRAS, or BRAF ablation in the parental MCF-10A, KRAS G12V/+, and BRAF V600E/+ cell lines grown in the partial growth medium. Three days after transfection with siRNA, relative cell densities were determined by crystal violet staining. (A) Representative 96-well plates are shown. (B) The mean relative cell number is shown with the SD (Parental, n = 4; KRAS G12V/+, n = 4; BRAF V600E/+, n = 5). Dots indicate actual data points. (C) Cell growth assays in the parental MCF-10A, KRAS G12V/+, and BRAF V600E/+ cell lines grown in starvation medium for four days with the indicated siRNA. Relative cell densities were determined by crystal violet staining. (Lower) Representative 96-well plates are shown. (Upper) Mean relative cell numbers are shown with the SD (Parental, n = 9; KRAS G12V/+, n = 22; BRAF V600E/+, n = 25). Dots indicate actual data points. (D and E) Cell growth assays following siRNA-mediated negative control, KRAS, or BRAF ablation in the KRAS G12V/+ and BRAF V600E/+ cell lines grown in starvation medium. Four days after transfection with siRNA, relative cell densities were determined by crystal violet staining. (D) Representative 96-well plates are shown. (E) The mean relative cell number is shown with the SD (KRAS G12V/+, n = 25; BRAF V600E/+, n = 25). Dots indicate actual data points. (F) Representative images of MTT-stained colonies of the indicated cells seeded in soft agar for 2 weeks. (G) The mean number of colonies (left) and the mean colony size ($\mu m^2$) (lower) are shown with the SD. Dots indicate actual data points. The numbers of experiments are as follows: KRAS G12V/+, n = 6; BRAF V600E/+, n = 6. * $p < 0.05$, ** $p < 0.01$.

same extent that the empty vector-introduced control cells did (Fig 5A). KRAS G12V OE cells displayed rapid cell growth in soft agar, forming large colonies one week after seeding (Fig 5B). Raptured spheroids were observed two weeks after seeding (Fig 5C), and finally, many small colonies were observed four weeks after seeding (Fig 5D and 5E). BRAF V600E OE late

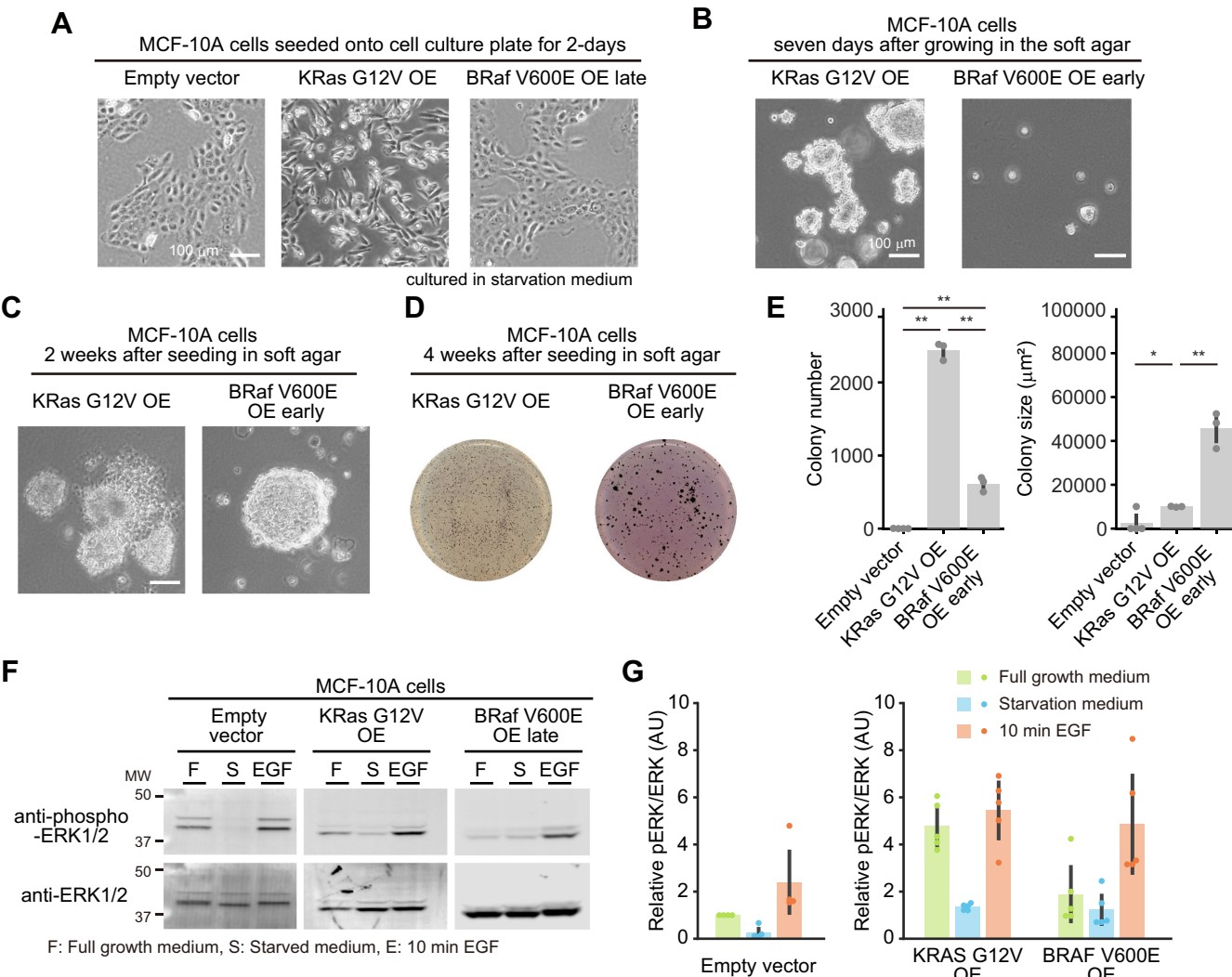

**Fig 5. Characterization of MCF-10A cells overexpressing KRAS G12V or BRAF V600E.** (A and B) Morphology of the indicated MCF-10A cells seeded onto a cell culture dish for two days (A) or seeded in soft agar for seven days (B). (C) Morphology of the indicated MCF-10A cells seeded in soft agar for 2 weeks. A white asterisk indicates the ruptured spheroid in KRAS G12V OE. (D) Representative images of MTT-stained colonies of the indicated MCF-10A cells seeded in soft agar for 4 weeks. (E) The mean number of colonies (left) and the mean colony size ($\mu m^2$)(right) are shown with the SD. Dots indicate actual data points. The numbers of experiments are as follows: Empty vector, n = 4; KRAS G12V OE, n = 4; BRAF V600E OE, n = 3. (F) Western blot analysis of the parental MCF-10A, KRAS G12V OE, and BRAF V600E OE late cell lines under the indicated conditions. (G) Quantification of ERK phosphorylation in panel F. Relative pERK/ERK values normalized by empty vector (left) and parental cells (right) under the full growth medium condition are shown with the SD (empty vector, n = 4; KRAS G12V OE, n = 5; BRAF V600E OE late, n = 5). Dots indicate actual data points. The results of statistical analysis are shown in S4B Fig. $p < 0.05$, ** $p < 0.01$.

cells showed no anchorage-independent growth in soft agar, but BRAF V600E OE early cells formed small colonies in soft agar after one week (Fig 5B), large colonies at two weeks that were comparable in size to those in anchorage-independent growth of BRAF V600E/+ cells for 4-weeks (S4C Fig), and finally, a small number of larger colonies at 4 weeks after seeding (Fig 5D and 5E). Consistent with the change in cell morphology and anchorage-independent growth, strong ERK phosphorylation was maintained in KRAS G12V OE cells under all conditions (Fig 5F and 5G). Despite the higher basal ERK phosphorylation in BRAF V600E OE early cells (S4A Fig), BRAF V600E OE late cells showed comparable or slightly higher levels of ERK phosphorylation in comparison to control cells (Fig 5F and 5G).

### Evaluation of oncogenic KRAS or BRAF addiction in MCF-10A cells overexpressing KRAS G12V or BRAF V600E

Finally, we quantified the effect of depletion of KRAS and BRAF with siPOOLs (siRNA) on cell growth in KRAS G12V OE and BRAF V600E OE late cells, respectively. The knockdown efficiency of siPOOLs targeting KRAS and BRAF was confirmed by western blotting (S5A and S5B Fig). To our surprise, KRAS or BRAF ablation only slightly reduced the cell growth rate in KRAS G12V OE or BRAF V600E OE late cells cultured in partial growth medium (Fig 6A and 6B). Of note, KRAS G12V OE cells have a spindle-like morphology (Fig 5A) that results in weaker crystal violet staining compared to other cells, but does not alter the rate of cell growth.

Like KRAS G12V/+ cells, KRAS G12V OE cells demonstrated growth factor independence (Fig 6C), and this was dependent on the expression of KRAS (Fig 6D and 6E, S5C and S5D Fig). BRAF V600E OE late cells showed a modest increase in cell growth under the serum starvation condition (Fig 6C). In addition, anchorage-independent growth in KRAS G12V OE cells were attenuated by KRAS depletion (Fig 6F and 6G). Taken together, the overexpression of KRAS G12V or BRAF V600E enhanced more or less *in vitro* transformation activity, but it did not suffice to induce oncogene addiction in MCF-10A cells.

## Discussion

In this study, we examined the association between oncogene addiction and *in vitro* tumorigenic properties (Fig 7A). In the human normal mammary gland-derived MCF-10A cell lines, an oncogenic mutation in a single allele of the *KRAS* or *BRAF* gene induced modest anchorage-independence, proliferative capacity, and phosphorylation of ERK, while the cells did not exhibit oncogene addiction. Similarly, MCF-10A cells overexpressing the oncogenic KRAS G12V or BRAF V600E protein demonstrated several properties of *in vitro* transformation properties, but did not show any signs of oncogene addiction. From these results, we conclude that, at least in the *KRAS* or *BRAF* gene of MCF-10A, the introduction of an oncogenic mutation or overexpression of an oncogene does not ensure the acquisition of oncogene addiction, and the properties of *in vitro* transformation are not necessarily coupled with oncogene addiction.

Why was the oncogenic mutation or over-expression of *KRAS* or *BRAF* in MCF-10A cells not sufficient to induce oncogene addiction? A reasonable possibility is that, after the introduction of the first oncogene mutation *in vivo*, tumor cells accumulate the oncogenes and/or tumor suppressor gene mutations with epigenetic alterations over a long period of time, gradually acquiring oncogene addiction (Fig 7B). Indeed, sensitivity to BRAF and MEK inhibitors, a feature of BRAF addiction, has been associated with distinct phenotype plasticity of the differentiation state and global alterations in gene expression programs in BRAF-mutated melanomas [34–36]. Further, it has been reported that a gene expression pattern associated with epithelial-mesenchymal transition (EMT) correlates with KRAS addiction [16]. Malignant tumor cells exhibiting the property of oncogene addiction may undergo environmental changes *in vivo* that render the cells oncogene-addicted. In other words, the *in vitro* culture of MCF-10A cells may be insufficient to cause genetic and/or epigenetic alteration leading to oncogene addiction. It would be of critical importance to identify such an environment or stimuli in order to enhance the effects of molecularly targeted drugs. Another (and not mutually exclusive) possibility is that oncogene addiction may be tissue-specific; thus mammary gland-derived MCF-10A cells may be inherently incapable of acquiring KRAS or BRAF addiction. Oncogenic KRAS or BRAF addiction has been found in lung and colon cancers and melanomas, where *KRAS* or *BRAF* is frequently mutated [16]. Human breast cancers rarely show *KRAS* and *BRAF* mutations, whereas mutations in genes that activate the PI3K-Akt pathway

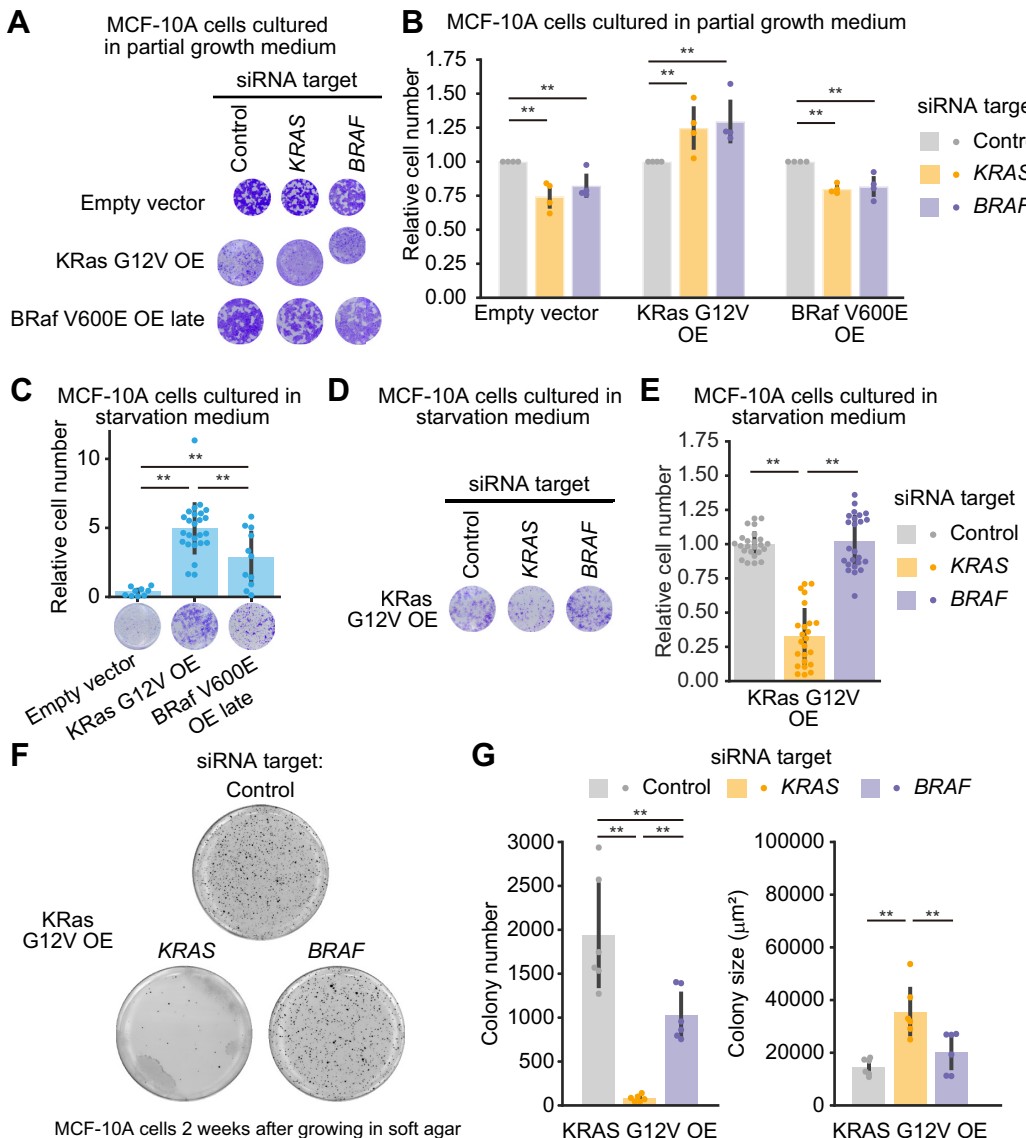

**Fig 6. Evaluation of oncogenic KRAS or BRAF addiction in MCF-10A cells overexpressing KRAS G12V or BRAF V600E.** (A and B) Cell growth assays following siRNA-mediated negative control, KRAS, or BRAF ablation in parental MCF-10A, KRAS G12V OE, and BRAF V600E OE late cell lines grown in the partial growth medium. Three days after transfection with siRNA, relative cell densities were determined by crystal violet staining. (A) Representative 96-well plates are shown. (B) The mean relative cell number is shown with the SD. The numbers of experiments are as follows: Parental, n = 4; KRAS G12V OE, n = 4; BRAF V600E OE late, n = 4. Dots indicate actual data points. (C) Cell growth assays in empty vector-introduced MCF-10A, KRAS G12V OE, and BRAF V600E OE late cell lines grown in the starvation medium for four days with the indicated siRNA. Relative cell densities were determined by crystal violet staining. (lower) Representative 96-well plates are shown. (upper) Mean relative cell numbers are shown with the SD (empty vector control, n = 9; KRAS G12V OE, n = 25; BRAFV600E OE late, n = 11). Dots indicate actual data points. (D and E) Cell growth assays following siRNA-mediated negative control, KRAS, or BRAF ablation in the KRAS G12V OE cell line grown in the starvation medium. Four days after transfection with siRNA, relative cell densities were determined by crystal violet staining. (D) Representative 96-well plates are shown. (E) The mean relative cell number is shown with the SD (Control, n = 25; KRAS, n = 24; BRAF, n = 24). Dots indicate actual data points. (F) Representative images of MTT-stained colonies of the indicated cells seeded in soft agar for 2 weeks. (G) The mean number of colonies (left) and the mean colony size ($\mu m^2$)(lower) are shown with the SD (KRAS G12V OE, n = 6). Dots indicate actual data points. * $p < 0.05$, ** $p < 0.01$.

**A**

| | In vitro tumorigenic properties | | | | | Oncogene addiction |
|---|---|---|---|---|---|---|
| | pERK | 2D morphology | Anchorage independence | | Growth factor independence | |
| Parental MCF-10A | - | Epithelial | - | | - | - |
| KRAS G12V/+ | +/- | Epithelial | + | KRAS dependent | ++ | KRAS dependent | - |
| BRAF V600E/+ | ++ | Fibroblastic | ++ | KRAS, BRAF dependent | +++ | BRAF independent | - |
| KRAS G12V OE | ++ | Fibroblastic | +++ | KRAS dependent | + | KRAS dependent | - |
| BRAF V600E OE | +/- | Epithelial | +/- | | +/- | | - |

**B**

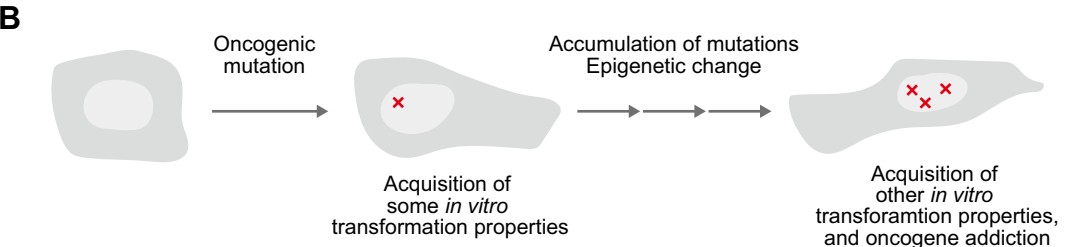

Oncogenic mutation

Accumulation of mutations
Epigenetic change

Acquisition of some *in vitro* transformation properties

Acquisition of other *in vitro* transforamtion properties, and oncogene addiction

**Fig 7. Model for the acquisition of oncogene addiction.** (A) A table summarizing the results of this study. (B) Schematic diagram showing how oncogene addiction is acquired.

have been reported [37]. Interestingly, Her2 amplification and PIK3CA$^{H1047R}$-positive breast cancer exhibited PI3K dependency in a mouse model [38], suggesting that genes that are strongly involved in tumorigenesis in a specific tissue are also likely to show oncogene addiction.

In this study, we observed phenotypic differences in MCF-10A cells expressing oncogenic KRAS or BRAF. These differences may have been due to differences in the expression levels and/or differences between KRAS G12V and BRAF V600E. Significant increases in ERK phosphorylation and tumorigenic activity were not observed in KRAS G12V/+ cells (Fig 2), whereas both changes were observed in KRAS G12V OE cells, which expressed oncogenic KRAS G12V at approximately 20-fold the level of endogenous KRAS (S1C and S1D Fig). Interestingly, neither case demonstrated an oncogene-induced senescence phenotype. It has been reported that cellular senescence is induced when the expression of HRAS G12V is twice that of endogenous RAS in MCF-10A cells and human mammary epithelial cells/hTERT [39]. The higher expression of KRAS G12V in mouse mammary epithelium *in vivo* has been found to induce cellular senescence in the p53-p16 pathway dependent manner [40]. However, homozygous loss of *p16* in MCF-10A cells has been demonstrated [41], which probably makes the cells immortal and resistant to cellular senescence caused by KRAS G12V overexpression. These results strongly suggest a dose-dependent switch between cellular senescence and tumorigenesis of KRAS G12V in mammary epithelial cells. With respect to BRAF, our results showed that the expression of BRAF V600E from the endogenous gene locus enhanced ERK phosphorylation and *in vitro* transformation activity (Fig 2), but these characteristic features were gradually diminished when BRAF V600E was overexpressed, and eventually, cells with low BRAF V600E expression were selected (Fig 5 and S4 Fig). Although BRAF V600E-induced cellular senescence has not been reported in mammary epithelial cells, melanocytes and

fibroblasts have been reported to show cellular senescence by overexpression of BRAF V600E [42]. It is reasonable to assume that overexpression of BRAF V600E induced cellular senescence, thereby leading to the selection of cells with low BRAF V600E expression. However, the susceptibility of oncogene-induced senescence is cell type-dependent. In fibroblasts, overexpression of oncogenic KRAS induces senescence [43], while endogenous KRAS G12D expression enhances cell proliferation [44]. In the future, a more quantitative investigation will be needed to reveal the relationship between the expression level of oncogenes and the consequence of tumorigenesis and cellular senescence.

Future studies should focus on the development of an experimental system for the acquisition of oncogene addiction in various cell lines including lung and pancreas-derived cells. It has been reported that overexpression of Myc followed by suppression leads to apoptosis as a model of oncogene addiction [45]. However, it is technically difficult to trace the process by which cells acquire oncogene addiction in an *in vivo* model. Therefore, it will be necessary to develop an *in vitro* experimental system for the acquisition of oncogene addiction. Understanding what environmental changes lead to oncogene addiction and what state changes the cell undergoes in the process will be important in augmenting the effects of molecularly targeted drugs.

## Supporting information

**S1 Fig. Characterization of MCF-10A cells harboring an oncogenic mutation in a single allele of KRAS or BRAF.** (A) Genome sequence of *KRAS* gene (left) and *BRAF* gene (right) in the parental MCF-10A, KRAS G12V/+, and BRAF V600E/+ cell lines. (B) The results of statistical analysis in Fig 2G are shown in the heatmap. See Materials and Methods for the details of the statistical analysis. (C-E) Expression levels of KRAS, BRAF, and ERK in MCF-10A cell lines used in this study. (C) Western blot analysis of the MCF-10A cell lines. (D) Quantification of the expression levels of KRAS, BRAF, and ERK in panel C. Relative values normalized by parental MCF-10A cells cultured in full growth medium condition are shown with the SD (n = 3 in all cell lines). Dots indicate actual data points. (E) The results of statistical analysis are shown in the heatmap. $^{*}$ $p < 0.05$, $^{**}$ $p < 0.01$.
(EPS)

**S2 Fig. Knock-down efficiencies in cancer-derived cell lines.** (A and B) Knock-down efficiencies of KRAS and BRAF using targeted siPOOLs in A549 cells, H358 cells, and A375 cells cultured in RPMI supplemented with 10% FBS were analyzed by western blotting. (A) The representative western blot images are shown. (B) Quantification of the expression levels of KRAS, BRAF, ERK, and pERK in panel A. Relative values normalized by the value of negative control are shown with the SD. The numbers of experiments are as follows: A549, n = 5; H358, n = 5; A375, n = 6. Dots indicate actual data points. (C and D) Knock-down efficiencies of KRAS and BRAF using targeted siPOOLs in H358 cells cultured in RPMI supplemented with 5% FBS were analyzed by western blotting. (C) The representative western blot images. (D) Quantification of the expression levels of KRAS, BRAF, ERK, and pERK in panel C. Relative values normalized by the value of negative control are shown with the SD (n = 4). Dots indicate actual data points. $^{*}$ $p < 0.05$, $^{**}$ $p < 0.01$.
(EPS)

**S3 Fig. Knock-down efficiencies in MCF-10A cells harboring an oncogenic mutation in a single allele of KRAS or BRAF.** (A and B) Knock-down efficiencies of KRAS and BRAF using targeted siPOOLs in the indicated MCF-10A cells cultured in the partial growth medium were analyzed by western blotting. (A) The representative western blot images are shown. (B)

Quantification of the expression levels of KRAS, BRAF, ERK, and pERK in panel A. Relative values normalized by the value of negative control are shown with the SD. The numbers of experiments are as follows: Parental, n = 3; KRAS G12V/+, n = 3; BRAF V600E/+, n = 3. Dots indicate actual data points. (C and D) Knock-down efficiencies of KRAS and BRAF using targeted siPOOLs in the indicated MCF-10A cells cultured in the starvation medium were analyzed by western blotting. (C) The representative western blot images are shown. (D) Quantification of the expression levels of KRAS, BRAF, ERK, and pERK in panel C. Relative values normalized by the value of negative control are shown with the SD. The numbers of experiments are as follows: KRAS G12V/+, n = 4; BRAF V600E/+, n = 3. Dots indicate actual data points. $^*$ $p < 0.05$, $^{**}$ $p < 0.01$.
(EPS)

**S4 Fig. Characterization of MCF-10A cells overexpressing KRAS G12V or BRAF V600E.** (A) BRAF and pERK levels in the indicated MCF-10A cell lines were analyzed by western blotting. The representative images are shown. (B) The results of statistical analysis in Fig 5G are shown in the heatmap. $^*$ $p < 0.05$, $^{**}$ $p < 0.01$.
(EPS)

**S5 Fig. Knockdown efficiencies in MCF-10A cells overexpressing KRAS G12V or BRAF V600E.** (A and B) Knock-down efficiencies of KRAS and BRAF using targeted siPOOLs in the indicated MCF-10A cells cultured in the partial growth medium were analyzed by western blotting. (A) The representative western blot images are shown. (B) Quantification of the expression levels of KRAS, BRAF, ERK, and pERK in panel A. Relative values normalized by the value of negative control are shown with the SD. The numbers of experiments are as follows: Empty vector, n = 3; KRAS G12V OE, n = 3; BRAF V600E OE late, n = 3. Dots indicate actual data points. (C and D) Knock-down efficiencies of KRAS and BRAF using targeted siPOOLs in KRAS G12V OE cells cultured in the starvation medium were analyzed by western blotting. (C) The representative western blot images are shown. (D) Quantification of the expression levels of KRAS, BRAF, ERK, and pERK in panel C. Relative values normalized by the value of negative control are shown with the SD (n = 3). Dots indicate actual data points. $^*$ $p < 0.05$, $^{**}$ $p < 0.01$.
(EPS)

**S1 Raw images.**
(PDF)

## Acknowledgments

We thank Yusuke Miyanari for the use of the bioruptor; Emi Ebine, and Kaori Onoda for their assistance; and all members of the K.A. laboratory for helpful discussions. This work was supported by the Functional Genomics Facility, NIBB Core Research Facilities.

## Author Contributions

**Conceptualization:** Chitose Oneyama, Kazuhiro Aoki.

**Data curation:** Reina E. Ito.

**Funding acquisition:** Kazuhiro Aoki.

**Investigation:** Reina E. Ito.

**Methodology:** Reina E. Ito.

**Project administration:** Chitose Oneyama, Kazuhiro Aoki.

**Supervision:** Chitose Oneyama, Kazuhiro Aoki.

**Validation:** Reina E. Ito.

**Visualization:** Reina E. Ito, Kazuhiro Aoki.

**Writing – original draft:** Reina E. Ito, Chitose Oneyama, Kazuhiro Aoki.

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
