## [Decision Letter · Decision Letter 0]

30 Oct 2020

PONE-D-20-30780

Oncogenic mutation or overexpression of oncogenic KRAS or BRAF is not sufficient to confer oncogene addiction.

PLOS ONE

Dear Dr. Aoki,

Thank you for submitting your manuscript to PLOS ONE. After careful consideration, we feel that it has merit but does not fully meet PLOS ONE’s publication criteria as it currently stands. Therefore, we invite you to submit a revised version of the manuscript that addresses the points raised during the review process.

We look forward to receiving your revised manuscript.

Kind regards,

Romi Gupta

Academic Editor

PLOS ONE

Journal Requirements:

"This work was supported by the Functional Genomics Facility, NIBB Core Research Facilities. K.A. is

supported by JST, CREST Grant No. JPMJCR1654; and by MEXT/JSPS KAKENHI Grant No.

16KT0069, 16H01425 “Resonance Bio”, 18H04754 “Resonance Bio”, 18H02444, and

19H05798."

"K.A. is supported by JST, CREST Grant No. JPMJCR1654; and by MEXT/JSPS KAKENHI Grant No. 16KT0069, 16H01425 “Resonance Bio”, 18H04754 “Resonance Bio”, 18H02444, and 19H05798.

Reviewers' comments:

Reviewer's Responses to Questions

**Comments to the Author**

1. Is the manuscript technically sound, and do the data support the conclusions?

Reviewer #1: Partly

Reviewer #2: Partly

2. Has the statistical analysis been performed appropriately and rigorously? 

Reviewer #1: No

Reviewer #2: N/A

3. Have the authors made all data underlying the findings in their manuscript fully available?

Reviewer #1: Yes

Reviewer #2: Yes

4. Is the manuscript presented in an intelligible fashion and written in standard English?

Reviewer #1: Yes

Reviewer #2: Yes

5. Review Comments to the Author

Reviewer #1: Here Ito and colleagues have submitted a manuscript to explore when cells become ‘oncogene addicted’. They use modified MCF10A cells as a surrogate for normal mammary epithelial cells expressing physiologic levels or elevated levels of oncogenic versions of KRAS (G12V) or BRAF (V600E). Physiologic levels were obtained via knock-in mutations and overexpression was attained through lentiviral transduction. They assessed cell number reduction as a surrogate for transformation caused by knockdown of the oncogene in question.

Concerns:

1. Assessment of oncogenicity:

For a tumour or cell line to be oncogenically addicted it must be shown to be oncogenic and that the removal of the oncogene then prevents oncogenicity. (a) the authors never show that these cells are capable of forming tumours in recipient animals. While soft agar assay positivity does correlate with tumor formation this is not always the case. (b) The bigger concern however, is that there is no use of soft agar assays (ie transformation assay) to determine whether the cells remain transformed after knockdown of the oncogene in question. This undoubtedly is because siRNA pools were used, which while being a standard first start to addressing a requirement, fail in any longer-term assessment of requirement. Longer term experiments can be accomplished through the use of lentiviral/retroviral vectors (even with inducible knockdown) that are completely compatible with longer term assays and make the cell population more uniform. Additionally, for BRAF the use of BRAFV600E specific drugs could have been employed in longer assays. Both approaches would work for tumorgenicity assays and soft agar transformation assays (if the authors do not have access to resources necessary for the mouse work). Without these, the only thing that the authors are able to study is a deviation in cell number and as proliferation in culture is not reflective of transformation or oncogenicity this makes statements about addiction difficult to assess. For example, while it is true that a severe reduction in proliferation would most likely cause a decrease in tumorigenicity, many proliferating cells are not themselves transformed - for instance the parental MCF10A cells are proliferative but not tumorigenic. Thus, if one uses proliferation as the sole assay for oncogene addiction then you are likely to mischaracterize phenotypes. Since the authors haven’t measured transformation as a readout then they really cannot say whether or not their cells are oncogene addicted. I encourage the authors to either assess this using lentiviral knockdown or alter the text considerably to reflect what they are assessing here (growth factor independence effects on proliferation).

2. Choice of cells for this study:

This reviewer is slightly concerned about the use of MCF10a cells as a surrogate of normalcy. Despite being widely used, MCF10A cells do have properties that distinguish them from primary mammary epithelial cells (immortality as well as genomic rearrangements Cancer Genet Cytogenet, 2005 Nov;163(1):23-9.). But this concern is eclipse by the concern that the authors chose to study the effects of KRAS/BRAF mutations in breast epithelial cells. A quick search of the TCGA reveals that in ~6800 breast tumors 62 patients had 62 mutations in one or both of these genes (ie approximately 1% of breast cancer samples). One wonders why there is a lack of RAS/RAF mutations. Are mammary cells simply resistant to hyperactivation of the RAS/RAF? Are they sensitive to such signals and unduce senescence? Indeed, there is data to support both notions (Nature Cell Biol volume 9,pages493–505 (2007)). The authors should state why they used this particular cell line for their studies in the text and in the discussion, what the caveats are for using such lines (vs normal mammary epithelial cells)

3. Data quality, presentation analysis:

a) The supplemental data supporting knockdown in the cancer derived lines and MCF10A-derivatives was rather weak. These are not very convincing blots (specifically those in Supp fig S2/S3). It is unclear how many times these were done. Quantification would be bolstered by repetition and statistics. In particular a direct comparison of MCF10A-derivatives should be presented side-by-side on one blot.

b) Are the blots within a given panel at the same levels of exposure? For example, figure 1D shows quantification relative to parental controls in full growth media but the blots are shown as being separate. In other situations (FgiS3a) the blots are exposed differently.

c) Figures involving protein quantification (2d, 4D) and cell number (3b,d,e,g5bce) should display that actual data points in addition to a graph with error bars) AND indicate statistical significance where present (with a statement or indicator where it is not). This would allow the reader to more accurately assess the data presented.

This reviewer was surprised by the statement that the MCF10A-derived cells grew too fast to “reproducibly quantify the effect of RNAi on KRAS or BRAF ablation”. Do these cells grow faster than A549s or the other tumor derived cell lines? If not, the issue would most likely be due to transfection efficiency.

4. Finally this reviewer is concerned about inconsistencies with the published literature. MCF10A cells with KRAS G12D knockin mutations have different properties than the cells used here (the authors point this out). Are these cells different or is the same source? (the ones used in this paper are purchased) If they are the same cells, then why would there be a difference? If they are different cells, then what could be an explanation that could reconcile these data? The use of siRNAs to KRAS do not replicate data seen by others (including Jeff Settelman’s group). Again, this is perplexing and without explanation.

5. Minor Text issues:

Line 134 reads, “…mutation of BRAF V600E in MCF-10A cells enhances in vitro tumorigenic activity… ”. The authors do not measure tumor formation, they are measuring transformation in culture. The text should be revised.

Figure 6 uses the word “tumorigenisities ”. I am not sure this is a word.

Reviewer #2: This paper by Ito et el. uses cell line systems to investigate the induction of oncogene addiction following endogenous or exogenous expression of mutant KRAS(G12V) or BRAF(V600E). The authors conclude that expression of single mutant oncogenes in defined in vitro cell culture models does not lead to oncogene addiction. Unfortunately, the data as presented do not support the major conclusion of the paper. Firstly, as the authors acknowledge (lines 250-251), KRAS or BRAF mutations are rarely found in breast cancer. Therefore, the use of MCF10A mammary epithelial cells as the only model for the experiments limits the ability to generalize conclusions. Additional cell line models should be tested using lung and pancreas-derived cell lines. Secondly, the authors constantly refer to “tumorigenicity”. This term refers the ability of transformed cells to form tumors in vivo. The in vitro experiments in this paper assess cellular transformation not tumorigenicity. These terms are distinct. The text should be changed accordingly. Another major concern is that the definition of oncogene addiction in this paper is vague. Soft agar assays are used to show that oncogene expression causes anchorage-independent growth. However, the effect of oncogene depletion is only tested under anchorage-dependent growth conditions. Will oncogene addiction be observed if KRAS and BRAF are depleted in cells grown in soft agar? This should be tested. Other concerns are noted below:

1. Figure 2B: why is the result inconsistent with published results? Please explain in more detail.

2. In all figures soft agar colony growth should be quantitated either as colony size or total colony number.,

3. Line 149-150: the reference to the A375 cell line as BRAF addicted is incorrect. Please change it.

4. Line 155: pERK does not change with siKRAS in H358 cells, which contradicts a previous report. It should be noted that H358 cells were grown in 10% FBS in this study, compared to 5% FBS for the Singh et al. study. Will the result change in lower serum conditions?

5. BRAF expression diminishes over time in MCF10A cells, which the authors state is an adaptation or negative feedback phenomenon. This is not a valid explanation. It could simply be a technical issue with the viral vector used to express BRAF, such as epigenetic silencing of the promoter. It is recommended to remove the data with late passage cells and include data from early passage cells to avoid confusion.

6. PLOS authors have the option to publish the peer review history of their article (what does this mean?). If published, this will include your full peer review and any attached files.

Reviewer #1: No

Reviewer #2: No

---

## [Author Response · Author response to Decision Letter 0]

1 Mar 2021

We are grateful to all reviewers for the critical comments and useful suggestions that have helped improve our paper. We provide point-to-point answers to the comments and added detailed explanations in the revised manuscript. We put the comments made by the reviewer in bold italics with our responses below. Our changes in the manuscript are marked with yellow highlights.

Review Comments to the Author

Reviewer #1: 

1. Assessment of oncogenicity:

For a tumour or cell line to be oncogenically addicted it must be shown to be oncogenic and that the removal of the oncogene then prevents oncogenicity. (a) the authors never show that these cells are capable of forming tumours in recipient animals. While soft agar assay positivity does correlate with tumor formation this is not always the case. (b) The bigger concern however, is that there is no use of soft agar assays (ie transformation assay) to determine whether the cells remain transformed after knockdown of the oncogene in question. This undoubtedly is because siRNA pools were used, which while being a standard first start to addressing a requirement, fail in any longer-term assessment of requirement. Longer term experiments can be accomplished through the use of lentiviral/retroviral vectors (even with inducible knockdown) that are completely compatible with longer term assays and make the cell population more uniform. Additionally, for BRAF the use of BRAFV600E specific drugs could have been employed in longer assays. Both approaches would work for tumorgenicity assays and soft agar transformation assays (if the authors do not have access to resources necessary for the mouse work). Without these, the only thing that the authors are able to study is a deviation in cell number and as proliferation in culture is not reflective of transformation or oncogenicity this makes statements about addiction difficult to assess. For example, while it is true that a severe reduction in proliferation would most likely cause a decrease in tumorigenicity, many proliferating cells are not themselves transformed - for instance the parental MCF10A cells are proliferative but not tumorigenic. Thus, if one uses proliferation as the sole assay for oncogene addiction then you are likely to mischaracterize phenotypes. Since the authors haven’t measured transformation as a readout then they really cannot say whether or not their cells are oncogene addicted. I encourage the authors to either assess this using lentiviral knockdown or alter the text considerably to reflect what they are assessing here (growth factor independence effects on proliferation).

We would like to appreciate the reviewer for his/her valuable comments. We agree with the critics pointed out by this reviewer, and therefore we have examined lentiviral-mediated knock-down of KRAS or BRAF and its effect on the soft agar colony formation assay during these three months. 

Initially, we tried lentiviral-mediated knock-down, using the same target sequence of shRNA for KRAS gene as used by Singh et al (please see the Methods in this rebuttal for details). 

We determined the titler of these lentiviruses (see Methods) and confirmed the effect of knock-down by western blotting (Figure 1 for reviewer only). The first problem was that shRNA-mediated knock-down of KRAS was not effective in MCF-10A/KRAS G12V OE cells (red arrows in Figure 1). This could be due to the high expression of KRAS in the cell line. 

Figure 1 for reviewer only. Knock-down of KRAS or BRAF through lentivirus-mediated shRNA. (A) The indicated cells seeded on 12 well plates were infected with lentivirus expressing scramble shRNA (sh-Scr), shRNA for KRAS gene (sh-KRAS-1), or shRNA for BRAF gene (sh-BRAF-1). One day after infection, their culture medium was replaced with a fresh culture medium with 2.0 μg/mL puromycin to remove uninfected cells. Two to three days after puromycin selection, Cell lysates were prepared with 1xSDS sample buffers of varying volumes so that the number of cells was comparable, and subjected to western blotting. (B) The KRAS, BRAF, and pERK levels are plotted as bar graphs in each cell line. These expression levels of KRAS, BRAF, and pERK1/2 are divided by ERK1/2, followed by the normalization by dividing by the value of sh-Scr. Of note, shRNA for KRAS gene had no impact on the knock-down of KRAS proteins in MCF-10A/KRAS G12V OE cells. 

The second problem was that all cell lines including MCF-10A/KRAS G12V OE cells gradually died after infection of lentivirus expressing shRNA for KRAS gene, even though this shRNA did not exhibit KRAS knock-down in MCF-10A/KRAS G12V OE cells (Figure 2 for reviewer only). This suggests the non-specific cytotoxicity of the lentivirus expressing shRNA for KRAS gene. We considered that the soft agar colony formation assay would not be suitable under these conditions.

Figure 2 for Reviewer only. Morphology of cells infected with the lentivirus expressing shRNAs. The cells were seeded in 6-well plate to a cell density of 9x10^4 cells/well (MCF-10A/KRAS G12V/+), 1.2x10^5 cells/well (MCF-10A/BRAF V600E/+), 1.0x10^5 cells/well (MCF-10A/KRAS G12V OE), and 1.2x10^5 cells/well (MCF-10A/BRAF V600E OE). One day after seeding, the cells were infected with the lentivirus expressing control shRNA (sh-Scr, scramble), shRNA for KRAS gene (sh-KRAS), or shRNA for BRAF gene (sh-BRAF) with polybrene. One day after infection, the medium was replaced with fresh medium + 2.0 μg/mL puromycin to select infected cells. Three days after selection, the cells were imaged with Olympus DP21 equipped with 4x objective lens.

For these reasons, we gave up long-term knock-down of KRAS or BRAF genes by lentivirus, and instead, we used siPOOLs to knock-down KRAS or BRAF, followed by soft agar colony formation assay. As this reviewer pointed out, the effect of siPOOLs-mediated knock-down does not last for a long time, and therefore we presume that siPOOLs-mediated knock-down affects the effect of cell survival and proliferation in the early stages of the soft agar colony formation. 

In cancer-derived cell lines, we found that the number of colonies in A549 cells (non-addicted) and H358 cells (KRAS-addicted) was significantly reduced by KRAS knock-down (Fig. 3C and 3D). On the other hand, in A358 cells (BRAF-addicted), the number of colonies decreased by the knock-down of BRAF (Fig. 3C and 3D). These results seem to be consistent with the results of the proliferation assay.

Because three MCF-10A cell lines including parental, empty vector, and BRAF V600E OE late (data not shown) were incapable of soft agar colony formation in the first place (Figs. 2B-2E and 5B-5E), we did not perform soft agar colony formation assay with knock-down experiments on these cells. Thus, we investigated the effect of KRAS or BRAF knock-down on soft agar colony formation in the other three MCF-10A cell lines including KRAS G12V/+ cells, BRAF V600E/+ cells, and KRAS G12V OE cells. As a result, KRAS knock-down highly reduced the number of colonies in all three cell lines, and BRAF knock-down also decreased the number of colonies in all three cell lines, especially in BRAF V600E/+ cells (Figs. 4F, 4G, 6F, and 6G). From these results, we could evaluate the roles of KRAS and BRAF in the tumorigenic activity of these cells (Fig. 7). These results have been included in the revised manuscript and figures (Figs. 3C, 3D, 4F, 4G, 6F, and 6G).

On the other hand, we consider that these results are not suitable for the evaluation of oncogene addiction. This is because control MCF-10A cells (parental and empty vector) do not proliferate in the soft agar colony formation assay. If oncogene addiction has been acquired, we expect that knock-down of the oncogene would result in phenotypes such as apoptosis in oncogene addicted cells, but not in normal cells and oncogene non-addicted cells. However, as with cell proliferation in starvation medium (Figs. 4C-4E, and 6C-6E), the soft agar colony formation assay does not meet this criterion because control cells do not proliferate in the first place, and therefore these results were not used to evaluate oncogene addiction.

2. Choice of cells for this study:

This reviewer is slightly concerned about the use of MCF10a cells as a surrogate of normalcy. Despite being widely used, MCF10A cells do have properties that distinguish them from primary mammary epithelial cells (immortality as well as genomic rearrangements Cancer Genet Cytogenet, 2005 Nov;163(1):23-9.). But this concern is eclipse by the concern that the authors chose to study the effects of KRAS/BRAF mutations in breast epithelial cells. A quick search of the TCGA reveals that in ~6800 breast tumors 62 patients had 62 mutations in one or both of these genes (ie approximately 1% of breast cancer samples). One wonders why there is a lack of RAS/RAF mutations. Are mammary cells simply resistant to hyperactivation of the RAS/RAF? Are they sensitive to such signals and unduce senescence? Indeed, there is data to support both notions (Nature Cell Biol volume 9, pages 493–505 (2007)). The authors should state why they used this particular cell line for their studies in the text and in the discussion, what the caveats are for using such lines (vs normal mammary epithelial cells).

We would like to thank the reviewer for his/her effort to investigate the RAS/RAF mutation in mammary epithelial cells, and reasonable suggestions. According to the suggestion, we have included the following explanations in the revised manuscript:

Page 9, line 230

“This is because MCF-10A was spontaneously immortalized without defined factors [26], the cell line is not tumorigenic, i.e., they are not able to grow under anchorage-independent conditions or to form tumors when injected subcutaneously into nude mice [27], it does not have mutations in KRAS and BRAF, and it is easy to culture. In addition, it is beneficial to use MCF-10A because of the availability of cell lines with KRAS G12V and BRAF V600E as we mention below. On the other hand, it has been reported that MCF10A lacks a tumor suppressor gene, p16, which may render the cell line immortalized. Therefore, it should be noted that it is substantially different from normal mammary epithelial cells (see Discussion).”

Page 24, line 531

“The higher expression of KRAS G12V in mouse mammary epithelium in vivo has been found to induce cellular senescence in the p53-p16 pathway dependent manner [40]. However, homozygous loss of p16 in MCF-10A cells has been demonstrated [41], which probably makes the cells immortal and resistant to cellular senescence caused by KRAS G12V overexpression. These results strongly suggest a dose-dependent switch between cellular senescence and tumorigenesis of KRAS G12V in mammary epithelial cells. “

3. Data quality, presentation analysis:

a) The supplemental data supporting knockdown in the cancer derived lines and MCF10A-derivatives was rather weak. These are not very convincing blots (specifically those in Supp fig S2/S3). It is unclear how many times these were done. Quantification would be bolstered by repetition and statistics. In particular a direct comparison of MCF10A-derivatives should be presented side-by-side on one blot.

We agree with the reviewer’s comment that we require to demonstrate the repetition and statistics in western blotting data. We have included individual data points in all experiments including western blotting. The sample number, experimental condition, and statistical significance are described in the legends of those figures. In addition, we have included the data showing a direct comparison of expression levels of KRAS, BRAF, and ERK in all cell lines used in this study in Supplementary Figure S1C-E.

b) Are the blots within a given panel at the same levels of exposure? For example, figure 1D shows quantification relative to parental controls in full growth media but the blots are shown as being separate. In other situations (FgiS3a) the blots are exposed differently.

In all western blotting experiments, we detect the signal by near-infrared fluorescence with Odyssey CLx Infrared Imaging System (LI-COR), not conventional chemiluminescence. Therefore, the intensity of the excitation light and the scanning speed during detection affect the intensity of the signal. The conditions for this experiment are described in Materials and methods as follows:

 Page 7, line 187

“The detection conditions are as follows: Resolution, 168 μm (Figs. 2F, 5F KRAS and BRAF, S3C, S4A, and S5C) or 84 μm (Figs. 5F empty vector, S1D, S2A, S2C, S3A, and S5A); sensitivity (scanning speed), normal (Figs. 2F, 5F KRAS and BRAF, S3C, S4A, and S5C) or low (Figs. 5F empty vector, S1D, S2A, S2C, S3A, and S5A).”

c) Figures involving protein quantification (2d, 4D) and cell number (3b,d,e,g5bce) should display that actual data points in addition to a graph with error bars) AND indicate statistical significance where present (with a statement or indicator where it is not). This would allow the reader to more accurately assess the data presented.

According to the reviewer’s suggestion, we have shown individual data points of all experiments and results of statistical tests in the revised figures. 

This reviewer was surprised by the statement that the MCF10A-derived cells grew too fast to “reproducibly quantify the effect of RNAi on KRAS or BRAF ablation”. Do these cells grow faster than A549s or the other tumor derived cell lines? If not, the issue would most likely be due to transfection efficiency.

We apologize for the reviewer’s confusion, and please allow us to add an explanation. The cell proliferation rate of MCF-10A cells tends to be strongly dependent on cell density. Therefore, the seeding density at the time of cell passage has a great impact on the assay even a few days after the passage. It was quite difficult to determine the seeding cell number; if it is too small, all cells will die, and if it is too large, the cells will be overconfluent. In addition, the strong cell-to-cell adhesion of MCF-10A cells makes it difficult to quantitatively count the number of cells at the time of passage. For these reasons, we have tried to increase the reproducibility by reducing the cell proliferation rate with the partial growth medium, and have successfully obtained data. We have added these important notes in the revised manuscript as follows:

 Page 14, line 340

“This is because the seeding density at the time of cell passage has a great impact on the assay even a few days after the passage.”

4. Finally this reviewer is concerned about inconsistencies with the published literature. MCF10A cells with KRAS G12D knockin mutations have different properties than the cells used here (the authors point this out). Are these cells different or is the same source? (the ones used in this paper are purchased) If they are the same cells, then why would there be a difference? If they are different cells, then what could be an explanation that could reconcile these data? The use of siRNAs to KRAS do not replicate data seen by others (including Jeff Settelman’s group). Again, this is perplexing and without explanation.

The MCF-10A/KRAS G12V/+ cells that we used were purchased from a commercially available source and are probably identical to the cells reported previously (Konishi, et al., 2007). Although the paper by Konishi et al. does not describe the details of the soft-agar colony formation assay experiment, it seems to be almost the same as our experimental conditions. The obvious differences are the number of seeding cells (2x10^4 cells vs. 3x10^4 cells), the interval between medium additions (every 3 days vs. every 1 week), and incubation time (4 weeks vs 3 weeks). We could not exclude the possibility that these differences might cause inconsistent results. Therefore, we have added these explanations in the revised manuscript as follows:

 Page 10, line 249

“This result is not consistent with the previous report using the same cells [28], possibly because of the difference of the experimental conditions under which the seeding cell number (3x10^4 cells vs 2x10^4 cells), the top layer agar concentration (0.3% vs 0.4%), the interval of medium addition (every 3 days vs every 1 week), and/or incubation time (4 weeks vs 3 weeks).”

The difference between our results and those of Jeff Settleman and colleagues, i.e., the lack of decrease in pERK by knock-down of KRAS gene in H358 cells, could be explained simply by the difference in experimental systems. We cultured H358 cells in RPMI supplemented with 10% FBS, while Singh et al. had cultured the cells in RPMI supplemented with 5% FBS, as pointed out by Reviewer #2. Indeed, we confirmed the reproducibility that ERK phosphorylation was reduced by KRAS knock-down in 5% FBS culture medium (Fig. S2C and 2D). We have included this result in the revised manuscript. 

 Page 12, line 299

“In A375 cells, knock-down of the BRAF gene caused a concomitant decrease in both pERK and cell proliferation, which was consistent with previous reports showing that cancer cells harboring BRAF V600E are addicted to ERK pathway (S2A and S2B Fig)[32]. In H358 cells cultured in RPMI supplemented with 10% FBS, knock-down of the KRAS gene suppressed cell proliferation but not pERK with statistical significance (S2A and S2B Fig). Meanwhile, we could reproduce that ERK phosphorylation was significantly reduced in H358 cells cultured in 5% FBS medium as previously shown (S2C and S2D Fig)[16].” 

5. Minor Text issues:

Line 134 reads, “…mutation of BRAF V600E in MCF-10A cells enhances in vitro tumorigenic activity… ”. The authors do not measure tumor formation, they are measuring transformation in culture. The text should be revised.

Figure 6 uses the word “tumorigenisities ”. I am not sure this is a word.

We have corrected these mistakes.

Reviewer #2: 

This paper by Ito et el. uses cell line systems to investigate the induction of oncogene addiction following endogenous or exogenous expression of mutant KRAS(G12V) or BRAF(V600E). The authors conclude that expression of single mutant oncogenes in defined in vitro cell culture models does not lead to oncogene addiction. 

We would like to thank the reviewers for the evaluations and critical comments. 

Unfortunately, the data as presented do not support the major conclusion of the paper. Firstly, as the authors acknowledge (lines 250-251), KRAS or BRAF mutations are rarely found in breast cancer. Therefore, the use of MCF10A mammary epithelial cells as the only model for the experiments limits the ability to generalize conclusions. Additional cell line models should be tested using lung and pancreas-derived cell lines. 

We agree with the reviewer’s comment, but it is unlikely that the same experiment could be performed in lung and pancreas-derived cell lines due to the time limitation. We would like to emphasize that this paper is a first step toward the understanding of molecular mechanisms of how cells acquire the property of oncogene addiction. 

Secondly, the authors constantly refer to “tumorigenicity”. This term refers the ability of transformed cells to form tumors in vivo. The in vitro experiments in this paper assess cellular transformation not tumorigenicity. These terms are distinct. The text should be changed accordingly. 

We agree with this comment, and according to the reviewer’s suggestion, we have replaced the term “tumorigenicity” with “transformation activity” throughout the revised manuscript. 

Another major concern is that the definition of oncogene addiction in this paper is vague. Soft agar assays are used to show that oncogene expression causes anchorage-independent growth. However, the effect of oncogene depletion is only tested under anchorage-dependent growth conditions. Will oncogene addiction be observed if KRAS and BRAF are depleted in cells grown in soft agar? This should be tested. 

We agree with the reviewer’s comment. This comment is exactly the same as Comment 1 raised by the reviewer 1. Please refer to the answer to Comment 1 from the reviewer 1. 

1. Figure 2B: why is the result inconsistent with published results? Please explain in more detail.

We agree with the comment, and have included the explanation as follows: 

 Page 10, line 249

“This result is not consistent with the previous report using the same cells [28], possibly because of the difference of the experimental conditions under which the seeding cell number (3x10^4 cells vs 2x10^4 cells), the top layer agar concentration (0.3% vs 0.4%), the interval of medium addition (every 3 days vs every 1 week), and/or incubation time (4 weeks vs 3 weeks).”

2. In all figures soft agar colony growth should be quantitated either as colony size or total colony number.

According to the reviewer’s suggestion, we have quantified the data obtained by soft-agar colony formation assay as described in Materials and Methods. Briefly, we quantified the number of colonies that exceeded a certain threshold size and the average size of colonies. These data have been included in the revised manuscript and figures (Figs. 2E, 3D, 4G, 5E, and 6G).

3. Line 149-150: the reference to the A375 cell line as BRAF addicted is incorrect. Please change it.

We would like to thank the reviewer. We have corrected the reference. 

4. Line 155: pERK does not change with siKRAS in H358 cells, which contradicts a previous report. It should be noted that H358 cells were grown in 10% FBS in this study, compared to 5% FBS for the Singh et al. study. Will the result change in lower serum conditions?

According to the reviewer’s suggestion, we examined the pERK level with siRNA for KRAS and BRAF in H358 cells cultured in 5% FBS as did in Singh et al. We could reproduce their results showing that siRNA for KRAS significantly reduced pERK level in H358 cells (Fig. S2C and S2D). We have included this result in the revised manuscript as follows:

 Page 12, line 299

“In H358 cells cultured in RPMI supplemented with 10% FBS, knock-down of the KRAS gene suppressed cell proliferation but not pERK with statistical significance (S2A and S2B Fig). Meanwhile, we could reproduce that ERK phosphorylation was significantly reduced in H358 cells cultured in 5% FBS medium as previously shown (S2C and S2D Fig)[16].”

5. BRAF expression diminishes over time in MCF10A cells, which the authors state is an adaptation or negative feedback phenomenon. This is not a valid explanation. It could simply be a technical issue with the viral vector used to express BRAF, such as epigenetic silencing of the promoter. It is recommended to remove the data with late passage cells and include data from early passage cells to avoid confusion.

As the reviewers mentioned, we agree that we could not exclude the possibility that the adaptation of BRAF signaling is attributed to the gene silencing when oncogenic BRAF is expressed for a long time. However, we decided not to omit these data because we considered it would be more beneficial to the readers and the scientific community to include them in the paper honestly. Instead, we have added the following explanation in the revised manuscript regarding the possibility of gene silencing as pointed out by the reviewer:

 Page 17, line 400

“During the course of experiments, we recognized that long-term culture of BRAF V600E OE cells reduced the expression levels of BRAF V600E and ERK phosphorylation levels (S4A Fig), probably due to the adaptation by a reduction of BRAF V600E expression through gene silencing and/or negative feedback mechanisms [33]”

Methods for Reviewers

Plasmids

pLKO.1-scr (scramble), pLKO.1-hKRAS, and pLKO-hBRAF. Plasmids were constructed in accordance with the standard molecular biology methods. The target sequences of the pLKO.1 vectors are as follows: scr, 5’-CCTAAGGTTAAGTCGCCCTCG-3’; hKRAS, 5’-CAGTTGAGACCTTCTAATTGG-3’ (Singh et al., 2009); and hBRAF, 5’-CCGGCAGCTTTCAGTCAGAT-3’. pLKO.1-TRC cloning vector was a gift from Dr. Root (Addgene plasmid # 10878). 

Lentivirus-mediated shRNA expression 

For delivery of shRNA via lentivirus, pLKO.1 vector was transfected into Lenti-X 293T cells (Clontech) together with the psPAX2, and pCMV-VSV-G-RSV-Rev by using the linear polyethyleneimine “Max” MW 40,000. Two days after transfection, the virus-containing supernatant was collected and filtered to remove 293T cells. The virus-containing supernatant was concentrated 20-fold by the Lenti-X concentrator (Clontech), and stored in a deep freezer. The cells were infected with the lentivirus in the presence of 8 μg/mL polybrene for 24 hrs, the virus-containing medium was replaced with a fresh culture medium with 2 μg/mL puromycin (InvivoGen), and the cells were further cultured for two or three days. The amount of virus to be infected was determined using HeLa cells, which showed no effect on KRAS or BRAF knock-down. Briefly, cells were seeded into 96 wells, and on the next day, the medium was replaced with the medium containing titrated lentivirus and 8 μg/mL polybrene. One day after infection, the virus-containing medium was changed to medium in the presence of 2 μg/mL puromycin. After two days, the number of survived cells was estimated by crystal violet staining. If the amount of virus was too high, the number of surviving cells decreased due to the virus toxicity, and if the amount of virus was too low, the number of infected cells decreased and most uninfected cells died by puromycin. Therefore, we determined the optimal amount of virus that showed the largest number of surviving cells.

---

## [Editor Report · Decision Letter 1]

18 Mar 2021

Oncogenic mutation or overexpression of oncogenic KRAS or BRAF is not sufficient to confer oncogene addiction.

PONE-D-20-30780R1

Dear Dr. Aoki,

We’re pleased to inform you that your manuscript has been judged scientifically suitable for publication and will be formally accepted for publication once it meets all outstanding technical requirements.

Kind regards,

Romi Gupta

Academic Editor

PLOS ONE
---

## [Editor Report · Acceptance letter]

22 Mar 2021

PONE-D-20-30780R1 

Oncogenic mutation or overexpression of oncogenic KRAS or BRAF is not sufficient to confer oncogene addiction. 

Dear Dr. Aoki:

I'm pleased to inform you that your manuscript has been deemed suitable for publication in PLOS ONE. Congratulations! Your manuscript is now with our production department. 

Kind regards, 

on behalf of

Dr. Romi Gupta 

Academic Editor

PLOS ONE